

# Conceptualizing community resilience to natural hazards – the emBRACE framework

Sylvia Kruse[1, 6,], Thomas Abeling[2], Hugh Deeming[3], Maureen Fordham[4], John Forrester[5], Sebastian Jülich[6], A. Nuray Karanci[7], Christian Kuhlicke[8], Mark Pelling[9], Lydia Pedoth[10], Stefan Schneiderbauer[10]

[1] Chair for Forest and Environmental Policy, University of Freiburg, 79106 Freiburg, Germany
[2] Climate Impacts and Adaptation, Environment Agency, 06844 Dessau-Roßlau, Germany
[3] HD Research, Lane Head, Bentham, N. Yorks, UK
[4] Department of Geography, Northumbria University, Ellison Place, Newcastle upon Tyne NE1 8ST, UK
[5] Stockholm Environment Institute & Environment Department, York Centre for Complex Systems Analysis, University of York, UK YO10
[6] Swiss Federal Institute for Forest Snow and Landscape Research, 8903 Birmensdorf, Switzerland
[7] Psychology Department, Middle East Technical University, Ankara, Turkey
[8] Helmholtz-Centre for Environmental Research – UFZ, 04318 Leipzig, Germany
[9] Department of Geography, King's College London, Strand, London WC2R, Great Britain
[10] Eurac Research, 39100 Bolzano, Italy

Correspondence to: Sylvia Kruse (sylvia.kruse@ifp.uni-freiburg.de)

**Abstract.** The level of community is considered to be vital for building disaster resilience. Yet, community resilience as a scientific concept often remains vaguely defined and lacks the guiding characteristics necessary for analysing and enhancing resilience on the ground. The emBRACE framework of community resilience presented in this paper provides a heuristic analytical tool for understanding, explaining and measuring community resilience to natural hazards. It was developed in an iterative process building on existing scholarly debates, on empirical case study work in five countries and on participatory consultation with community stakeholders, where the framework was applied and ground-tested in different contexts and for different hazard types. The framework conceptualizes resilience across three core domains: resources and capacities; actions; and learning. These three domains are conceptualized as intrinsically conjoined within a whole. Community resilience is influenced by these integral elements as well as by extra-community forces, comprising disaster risk governance and thus laws, policies and responsibilities on the one hand and on the other, the general societal context, natural and human-made disturbances and system change over time. The framework is a graphically rendered heuristic, which through application can assist in guiding the assessment of community resilience in a systematic way and identifying key drivers and barriers of resilience that affect any particular hazard-exposed community.

## 1 Introduction

Community resilience has become an important concept for characterizing and measuring the abilities of populations to anticipate, absorb, accommodate, or recover from the effects of a hazardous event in a timely and efficient manner (Almedom, 2013; Berkes and Ross, 2013; Deeming et al., 2014; Walker and Westley, 2011, 2011). This goes beyond a purely social-





ecological systems understanding of resilience (e. g. Armitage et al., 2012: 9) by incorporating social subjective factors, e.g. perceptions and beliefs, as well as the wider institutional environment and governance setting shaping the capacities of community to build resilience (Ensor and Harvey, 2015; Paton, 2005; Tobin, 1999). Many conceptual and empirical studies have shown that communities are an important scale and site for building resilience that can enhance both individual/household

and wider population level outcomes (Berkes et al., 1998; Cote and Nightingale, 2012; Nelson et al., 2007; Ross and Berkes, 2014).

Yet the community remains poorly theorised with little guidance on how to measure resilience building processes and outcomes. Both terms – resilience and community – incorporate an inherent vagueness combined with a positive linguistic bias, and are used with increasing frequency both on their own as well as in combination (Mulligan et al., 2016; Brand and

Jax, 2007; Strunz, 2012; Fekete et al., 2014). Both terms raise, as Norris et al. (2008) put it, the same concerns with variations in meaning. We broadly follow the definition of resilience proposed by the Fifth Assessment Report of the Intergovernmental Panel on Climate Change (IPCC): the capacity of social, economic, and environmental systems to cope with a hazardous event, trends or disturbance, responding or reorganizing in ways that maintain their essential function, identity, and structure, while also maintaining the capacity for adaptation, learning, and transformation (IPCC, 2014: 5).

In resilience research we can detect a disparity whereby the focus of research has often lain at either the larger geographical scales (e.g. regions), or, as in psychological research, it is focused at the level of the individual extending to households (Ross and Berkes, 2014; Paton, 2005). Across these scales and sites of interest resilience is consistently understood as relational. It is an ever emergent property of social-ecological and technological systems coproduced with individuals and their imaginations. As a relational feature, resilience is both held in and produced through social interactions. Arguably, most intense

and of direct relevance to those at risk, are such interactions at the local level including the influence of non-local actors and institutions. It is in this space that the 'community' becomes integral to resilience and an crucial level of analysis for resilience research (Cutter et al., 2008; Walker and Westley, 2011; Schneidebauer and Ehrlich, 2006).

The idea of community comprises groups of actors (e.g. individuals, organizations, businesses) which share a common identity. Communities can have a spatial expression with geographic boundaries with a common identity; or "shared fate" (Norris et

al., 2008: 128). Following the approach of Mulligan et al. (2016) we propose to apply a dynamic and multi-layered understanding of community including community as a place-based concept (e.g. inhabitants of a flooded neighbourhood); as a virtual and communicative community within a spatially extended network (e.g. members of crisis management in a region); and/or as an imagined community of individuals who may never have contact with each other, but who share an identity.

Considering the conceptual vagueness and variations of community and resilience, only a few approaches have tried to

characterize and measure community resilience comprehensively (Cutter et al., 2014; Sherrieb et al., 2010; Mulligan et al., 2016). Thus, the aim of this paper is to further fill this gap and elaborate a coherent conceptual framework for the characterization and evaluation of community resilience to natural hazards by building both on a top-down systems understanding of resilience and on an empirical, bottom-up perspective specifically including the 'subjective variables' and how they link to broader governance settings. The framework has been developed within the European research project





*emBRACE* in an iterative process building on existing scholarly debates, on empirical case study research in five countries (Great Britain, Germany, Italy, Switzerland, Turkey) using participatory consultation with community stakeholders, where the framework was applied and ground-tested in different regional and cultural contexts and for different hazard types. Further the framework served as a basis for guiding the assessment of community resilience on the ground.

The paper is structured as follows: the next section provides an overview of key themes and characteristics of conceptual frameworks on community resilience and identifies gaps and open questions in the current conceptual framings in the context of natural hazards. In section three we present the methodology for the development of the emBRACE framework of community resilience. In section four the emBRACE framework is introduced along its central elements and characteristics and illustrated by examples from the case study research. Section five discusses the interlinkages between the framework

elements as well as the application and operationalization of the framework and reflects on the results, methodology and further research.

## 2 Conceptual tensions of community resilience in disaster research and policy

One of the tensions surrounding the concept of resilience in the context of disaster risk reduction concerns its relation to social change and transformation. A divide is emerging between those that propose resilience as an opportunity for social reform and

transformation in the context of uncertainty (Bahadur and Tanner, 2014; Brown, 2014; Olsson et al., 2014; Kelman et al., 2015; MacKinnon and Derickson, 2013; Weichselgartner and Kelman, 2015), and those that argue for a restriction of the term to functional resistance and stability (Smith and Stirling, 2010; Klein et al., 2003).

Besides the differences in scope of the definition between bouncing back and societal change, there is another tension about whether resilience is a normative or an analytical concept (Fekete et al., 2014; Mulligan et al., 2016). The normative dimension

of resilience refers to its application as a policy instrument to promote disaster risk reduction at all scales (United Nations Office for Disaster Risk Reduction, 2015, 2007). The analytical dimension of resilience refers to its application as a lens to assess, evaluate, and identify options for building resilience (Cutter et al., 2008; Norris et al., 2008; Tyler and Moench, 2012). Both dimensions are often not distinct from each other, but rather overlap and are substantially intertwined. Many of the tensions around whether resilience is about social change, learning, and innovation can be attributed to this close integration

of normative and analytical aspects related to disaster resilience. Community resilience is not just a theoretical concept, but its use and application in disaster risk reduction policy has implications well beyond academic debates on climate change, adaptation, and disaster risk. Resilience is an integral element at the international policy level to both, the Hyogo Framework for Action and the Sendai Framework for Disaster Risk Reduction (United Nations Office for Disaster Risk Reduction, 2015, 2007) as well as to national and local discourses on disaster risk reduction, e.g. in the UK National Community Resilience

Programme (National Acadamies, 2012) or on the level of local authorities in the UK (Shaw, 2012).

We argue that the term community resilience is quickly acquiring prominence in disaster risk management policy-making across all scales, and is becoming part of political as well as academic discourses. Although in the context of natural hazards,





community resilience is often framed with a positive connotation, resilience-based risk reduction policy inevitably produces winners and losers (Bahadur and Tanner, 2014). In the UK, for example, resilience is part of a responsibilisation agenda in which responsibility for disaster risk reduction is intentionally devolved from the national to the local level (Department for Environment, Food and Rural Affairs, 2011; Deeming et al., 2017). This creates opportunities, but is also contested and can

provoke resistance by activists (Begg et al., 2016).

This normative dimension of community resilience and its relation to politics requires light being shed on the role of power and the distribution of responsibilities when analysing community resilience.

In this context, resilience is "here to stay" (Norris et al., 2008: 128) not only as a theoretical concept, but also as a policy tool to promote disaster risk reduction. As such it has direct implications for hazard prone communities. Debates about whether

resilience policy and practice should be limited to describe stability oriented aspects of disaster risk reduction (DRR), whilst leaving learning and social change for other concepts such as transformation, ignore the realities of disaster risk reduction action at the community level.

This importance of resilience "on the ground" has implications for the development and advancement of resilience theories. Frameworks of disaster resilience need to account for multiple entwined pressures, (e.g. development processes, DRR and

climate change cf. Kelman et al., 2015) to learn and adapt, and to innovate existing risk management regimes. Limiting resilience to narrow interpretations of robust infrastructure would promote local disaster risk reduction that fails to address the need for social change and reform, although these are proposed as being of critical importance to address uncertainties in the context of climate change (Adger et al., 2009).

Based on these arguments, we identify three gaps, in particular, that characterize existing resilience frameworks. First, there

seems to be an insufficient consideration and reflection of the role of power, governance, and political interests in resilience research. Secondly, many resilience frameworks still seem to fall short of exploring how resilience is shaped by the interaction of resources, actions, and learning. Due to the conceptual influence of the Sustainable Livelihood Framework (SLF) of some approaches (Chambers and Conway, 1992; Scoones, 1998; Ashley and Carney, 1999; Baumann and Sinha, 2001), resilience concepts tend to be focused on resources, but fail to systematically explore the interaction of resources with actions and

learning and how understanding these variables might then usefully illustrate disparities in how social equity, capacity and sustainability (i.e. key considerations of the SLF approach cf. Chambers and Conway, 1992) are manifest. Third, an explicit elaboration of learning and change is largely absent in the literature that characterizes community resilience. So far, resilience as a theory of change seems to remain rather vaguely specified.

A resilience framework which accounts for these aspects is necessarily focused on the prospects of social reform, and

incorporates many "soft" elements that are notoriously difficult to measure. We thus agree with the need to operationalize resilience frameworks (Carpenter et al., 2001), but argue that existing framework measurements (e.g. Cutter et al., 2008) often fail to systematically include those social aspects that we consider of critical importance for community resilience.



## 3 Framework development and methods used

Developing an interdisciplinary, multi-level and multi-hazard framework for characterizing and measuring resilience of European communities calls for the application of a multifaceted approach that adopts interdisciplinary methodological processes. Therefore, we applied a complementary research strategy, with the purpose of investigating resilience at different scales, from different perspectives and applying different research methods, as well as integrating the viewpoints of distinct actors.

A first strand of this research strategy included intensive structured literature reviews. The first sketch of the community resilience framework was informed by the early review systematizing the different disciplinary discussions on resilience into thematic areas. As the project continued, specialized literature reviews complemented this first review by focusing on different aspects of the framework and considering more recent publications. Throughout the project, developments in the literature were closely monitored and literature reviews were continuously updated (Abeling et al., 2017).

A second strand involved empirical case-study research in five European countries investigating community resilience related to different hazard types at different scales. The five case studies comprised multiple Alpine hazards in South Tyrol, Italy and Grisons, Switzerland, earthquakes in Turkey, river floods in Central Europe, combined fluvial and pluvial floods in northern England, and heatwaves in London. A number of qualitative and quantitative methodologies were adopted in the case study research in order to scrutinize the community resilience framework. The outcomes of this research have been used to inform the conceptual framework at different stages of the development process and helped to illustrate how the framework can be applied and adapted to different hazard types, scales and socio-economic and political contexts (Kuhlicke et al., 2016; Doğulu et al., 2016; Ikizer et al., 2015; Ikizer, 2014; Abeling, 2015b, 2015a; Taylor et al., 2014; Deeming H. et al., 2017; Jülich, 2017b, 2017a).

A third strand saw three participatory workshops with stakeholders in case studies in Cumbria, England; Van, Turkey; and Saxony, Germany in order to add to the framework development the perspective of different community stakeholders on the local and regional scale. The aim for the participatory assessment workshops was to collect, validate and assess the local appropriateness and relevance of different dimensions of community resilience and indicators to measure them. With the selection of case studies in different countries and different types of communities, we took into account that different cultures and communities conceptualize and articulate resilience differently. The workshops allowed discussion with local and regional stakeholders about how resilience can be assessed. This was both a presentation and revalidation of the first results of the case study work together with the stakeholders and also a starting point for further development of the framework.

A fourth strand involved internal review processes with project partners as well as external experts on community resilience.

## 4 The emBRACE framework for characterizing community resilience

The emBRACE framework conceptualizes community resilience as a set of intertwined components in a three-layer framework. First, the core of community resilience comprises three interrelated domains that shape resilience within the



community: resources and capacities; actions; and learning (cf. section 4.1). These three domains are intrinsically conjoint. Further, these domains are embedded in two layers of extra-community processes and structures (cf. section 4.2): first, in disaster risk governance which refers to laws, policies and responsibilities of different actors on multiple governance levels beyond the community level. It enables and supports regional, national and international civil protection practices and disaster

risk management organisations. The second layer of extra-community processes and structures is influenced by broader social, economic, political and environmental context factors, by rapid or incremental socio-economic changes of these factors over time and by disturbance. Together, the three-layers constitute the heuristic framework of community resilience (cf. figure 1), which through application can assist in defining the key drivers and barriers of resilience that affect any particular community within a hazard-exposed population.

**4.1 Intra-community domains of resilience: resources & capacities, action and learning**

**4.1.1 Resources & capacities**

The capacities and resources of the community and its members constitute the first domain of the core of resilience within the community. Informed through the Sustainable Livelihoods Approach (SLA) and its iterations (Chambers and Conway, 1992; Scoones, 1998; Ashley and Carney, 1999; Baumann and Sinha, 2001) as well as the concept of adaptive capacities (Pelling,

2011) we differentiate five types of capacities and resources. We believe that this approach also addresses in parallel the need identified by Armitage et al. (2012), for 'material', 'relational', and 'subjective' variables as well as the social subjective dimension of resilience (cf. section 1).

Natural and place-based capacities and resources relate to the protection and development of ecosystem services. This includes, but is not limited to, the role of land, water, forests and fisheries, both in terms of their availability for exploitation as well as

more indirectly for personal wellbeing of community members. Place-based resources can also refer to cultural and/or heritage resources, to local public services, amenities, and to the availability of access to jobs and markets.

Socio-political capacities and resources account for the importance of political, social and power dynamics and the capacity of community members to influence political decision-making. Here, institutions such as the rule of law, political participation and accountability of government actors are of critical importance. Participation in governance can be both formal, for example

through elections, and informal, for example through interest representation in political decision-making. Structural social resources are also inhered within the structural and cognitive components of social capital (Moser and McIlwaine, 2001), i.e. networks and trust. Social capital refers to lateral relationships between family, friends and informal networks, but also to more formal membership in groups, which may involve aspects of institutionalisation and hierarchy. Cognitively defined trust relationships can assist in collective action and knowledge-sharing, and thus seem integral for the development and

maintenance of community resilience (Longstaff and Yang, 2008). Operating within the framework's disaster risk governance domain, however, it should be acknowledged that mutual social-trust relations – as might be expressed between community





members, can be differentiated from 'trust in authority' wherein hierarchical power differentials introduce an element of dependency to the relationship (Szerszynski, 1999).

Financial capacities and resources refer to monetary aspects of disaster resilience. This includes earned income, pensions, savings, credit facilities, benefits, and importantly access to insurance. The role of financial capacities raises questions about

availability of and access to individual and public assets, and about the distribution of wealth across social collectives. The causal relationships that underpin the role of financial resources for community resources are not linear. Increases in available financial resources are not necessarily beneficial for community resilience, for example if income inequality is high and financial resources are concentrated in a very small and particular segment of society.

Physical capacities and resources for community resilience include adequate housing, roads, water and sanitation systems,

effective transport, communications and other infrastructure systems. This can also refer to the availability of and the access to premises and equipment for employment and for structural hazard mitigation (i.e. both at household and community scales). Finally, human capacities and resources focus at the individual level, integrating considerations such as gender, health and wellbeing, education and skills and other factors affecting subjectivities. Psychological factors are also accounted for here, with factors such as self-efficacy, belonging, previous hazard experience, coping capacities and awareness included. These

factors together can be understood to impact on both individuals' perceptions of risk and resilience but also as enablers of the community-based leadership that drives collective action.

From the case study in Turkey, socio-political (e.g., having good governance, specific disaster legislation, supervision of the implementation of legislation, coordination and cooperation, being a civic society, having mutual trust, having moral and cultural traditional values, etc.) and human (e.g., gender, income, education, personality characteristics, etc.) resources and

capacities were the most pronounced ones obtained (Karanci et al., 2017).

One of the participatory workshops where an earlier version of the framework was discussed with local stakeholders, in the case study on flooding in Northern England, revealed that for the participants' social-political as well as human capacities and resources were most important for characterizing their community resilience. Indicators measuring for example out-migration and in-migration as well as willingness to stay in the region and engage in associational activities were proposed to describe

the degree of community spirit and solidarity that was considered to be crucial for their community resilience in a region that is threatened by population loss and demographic change.

### 4.1.2 Actions

Within the emBRACE framework, community resilience comprises two types of actions: civil protection and social protection. The civil protection actions refer to the phases of the disaster management cycle, i.e. preparedness, response, recovery and

mitigation (Alexander, 2005). Resilience actions undertaken by the community can be related to these phases (e.g. weather forecasting and warning as preparedness action). Accordingly, civil protection is focusing on hazard specific actions. We add to this social protection considerations, which include hazard independent resilience actions, e.g. measures of vulnerability reduction and building social safety nets (cf. figure 1). Social protection action includes diverse types of actions intended to





provide community members with the resources necessary to improve their living standards to a point at which they are no longer dependent upon external sources of assistance (Davies et al., 2008). Social protection has been included as a main component because many resilience building actions cannot be directly attributed to civil protection action but are, rather, concerned with the more general pursuit of wellbeing and sustainability (Davies et al., 2013; Heltberg et al., 2009). For

example, the presence of an active community-based voluntary and/or charity sector capable of providing social support (e.g. foodbanks) and funding for participatory community endeavours (e.g. a community fund), and which could be extended or expanded in times of acute, disaster-induced, community need were found to be factors that provide a certain level of security for all those affected by hazards, either directly or indirectly (Dynes, 2005).

Such social protection measures are not, however delivered solely by the community and voluntary sector alone, so it is
important to understand that these elements also relate to the much broader provision of welfare services (health, education, housing, etc.), which are ultimately the responsibility of national and local government. The inclusion of social protection as a main component of this domain, therefore, represents an important progression over some other frameworks, because it explicitly includes the consideration of how communities manifest resilience through both, their capacity to deal with and adapt to natural hazards, but also their capacity to contribute equitably to reducing the wider livelihood-based risks faced by
some, if not all, of their membership.

In a case study in Northern England, social support mechanisms were particularly important across multiple communities (from hill farmers to town dwellers) in the aftermath of a flood event (Deeming et al., 2017). Key considerations were that despite evidence of learning and adaption that had occurred between two floods in 2005 and 2009, the sheer magnitude of the latter event effectively discounted the effects of any physical mitigation and civil protection measures that had been introduced.
Where non-structural measures, such as community emergency planning, had been adopted there were significant improvements in the levels and success of response activity. However, whilst these actions reduced some consequences (e.g. fewer vehicles flooded), where properties were inundated significant damage still resulted.  Accordingly, the importance of emergent community champions who were capable of advocating community outcomes, and the need for community spaces (e.g. groups or buildings), where those affected could learn by sharing experiences and deliberating plans, proved key factors
in driving the recovery, as well as the concurrently occurring future mitigation efforts. The fact that much of the support in the aftermath of the flood events was coordinated by particular officers from the statutory authorities, whose 'normal' roles and skills were social rather than civil protection orientated, itself emphasised the importance of understanding resilience in framework terms, as a practice-encompassing process rather than as a simple measure of hazard response capability.

### 4.1.3 Learning

Learning is the third integral domain that shapes intra-community resilience in the emBRACE framework. We attempt to provide a detailed conceptualization of learning in the context of community resilience. We follow the notion of social learning that may lead to a number of social outcomes, acquired skills and knowledge building, via collective and communicative learning (Muro and Jeffrey, 2008). It occurs formally and informally, often in natural and unforced settings via conversation



and mutual interest. Further, social learning is said to be most successful when the practice is spread from person to person (Reed et al., 2010) and embedded in social networks (McCarthy et al., 2011). In this understanding social learning is an on-going, adaptive process of knowledge creation that is scaled-up from individuals through social interactions fostered by critical reflection and the synthesis of a variety of knowledge types that result in changes to social structures (e.g. organizational mandates, policies, social norms) (Matyas and Pelling, 2015). Based on this understanding we conceptualise social learning as consistent of different elements from the perception of risks or losses, its problematisation, to the critical reflection and testing/experimentation in order to evolve new knowledge which can be disseminated throughout and beyond the community enabling resilience to embed at a range of societal levels (see figure 1). The first element, risk and loss perception grasps the ability of any actor, organisation or institution to have awareness of future disaster risk or to feel the impact of a current or past hazard event. Awareness can be derived from scientific or other forms of knowledge.

Second, the ability to problematise risk and loss arises once a threshold of risk tolerance is passed. A problematisation of risk manifests itself as a perception of an actor that potential or actual disaster losses, or the current achieved benefit to cost ratio of risk management are inappropriate. This includes procedural and distributional justice concerns, and has the potential to generate momentum for change. Third, critical reflection on the appropriateness of technology, values and governance frames can lead to a questioning of the risk-related social contract of the community. Critical reflection is proposed as a mechanism through which to make sense of what is being learned before applying it to thinking or actions.

Fourth, experimentation and innovation refers to the testing of multiple approaches to solving a risk management problem in the knowledge that these will have variable individual levels of success. This can shift risk management to a new efficiency mode where experimentation is part of the short-term cost of resilience and of long-term risk reduction. In this context, innovation can be conceptualised as processes that derive an original proposition for a risk management intervention. This can include the importing of knowledge from other places or policy areas as well as advances based on new information and knowledge generation.

Fifth, dissemination is integral for spreading ideas, practices, tools, techniques and values that have proven to meet risk management objectives across social and policy communities. Sixth and finally, monitoring and review refers to the existence of processes and capacity that can monitor the appropriateness of existing risk management regimes in anticipation of changing social and technological, environmental, policy, and hazard and risk perception contexts. The Turkish Case Study on earthquakes revealed that an earthquake experience in one region of the country led to learning mostly by the state and change and the adoption of new legislation and new organization for disaster management. Such an experience seems to have very robust effects on attitudes towards disasters, changing the focus from disaster management to disaster risk management (Balamir, 2002). The same change process seemed to apply to individuals as well, although to a smaller extent, in that an earthquake experience led to an increase in hazard awareness and preparedness.

The Italian Case Study in the Alpine village of Badia focuses on the perception of risks and losses as one element of resilience learning. The findings reveal that even though people living in Badia have high risk awareness, many did not expect and prepare for a manifesting event. The interpretation of the different risk behaviour profiles shows that people who perceive





themselves under risk of future landslide events had either personally experienced a landslide event in the past or participated in the clean-up work after the landslide event in 2012. Results of comparing the two groups of inhabitants affected by the landslide event 2012 and not affected in 2012 point in the same direction, showing that personal experience, not only recently but also if in the past, together with active involvement in the response phase lead to a higher risk perception especially when

thinking about the future (Pedoth et al., 2017).

## 4.2 Extra-community framing of community resilience

### 4.2.1 Disaster Risk Governance

In the proposed characterization of community resilience with respect to natural hazards, the three core domains – resources & capacities, actions and learning – are embedded in two extra-community frames. The first frame is that of formal and

informal disaster risk governance, which comprises laws, policies and responsibilities of disaster risk management at the local, regional, national and supra-national level. From the case study research it became clear that community resilience and its constituent resources and capacities, action and learning processes are strongly interacting with existing formal and informal laws, policies and responsibilities of civil protection and risk management more generally (e.g. flood mapping as per the German National Water Act and the EU Flood Directive). Responsibilities relates to the actors and stakeholders involved in

disaster risk management.

Relating the wider ideas of risk governance to the specific context of a community involves focus on the interaction between communities' resources and capacities, and actions as well as their learning processes to the specific framework by which responsibilities, modes of interaction and ways to participate in decision-making processes in disaster risk management are spelt out. The responsibilisation agendas in the two case studies in Cumbria, England and Saxony, Germany may serve as an

example. In both case studies community actions are being influenced by the downward-pressing responsibilisation agenda, which is encompassed for example within Defra's 'Making Space for Water' Strategy for Great Britain and Saxony's Water Law in Germany, the latter of which obliges citizens to implement mitigation measures. This explicitly parallels Walker and Westley's call to "push power down to the local community level where sense-making, self-organization, and leadership in the face of disaster were more likely to occur if local governments felt accountable for their own responses" (2011: 4). The

case study work showed that this not only relates to local governments (Begg et al., 2015; Kuhlicke et al., 2016) but also to the individual citizens potentially affected by natural hazards (Begg et al., 2016). More specifically, Begg et al. (2016) found that if the physical and psychological consequences are perceived as being low with regard to their most recent flood experiences, then respondents tend to accept the attribution of responsibility towards individual citizens and also report higher response efficacy (i.e. the respondents have the feeling they can reduce flood risk through their own actions) if they have taken

personal mitigation measures prior to the flood event. In addition, respondents who have taken personal mitigation measures are more likely than those who have not taken such actions to report higher response efficacy and also agree with the responsibility attributed to them. In other words, if respondents took personal mitigation measures before the flood and did not



experience severe consequences as a result of the flood, they are likely to agree with statements which support citizen responsibility and report high response efficacy. This shows that resilience action and learning processes are always embedded in the broader formal and informal risk governance settings.

### 4.2.2 Non-directly-hazard related context, social-ecological change and disturbances

As a second extra-community framing we consider three dimensions as influential boundary conditions for community resilience: first the social, economic, political and environmental context; second, social, economic, political and environmental change over time; and third diverse types of disturbances.

The first dimension of non-hazard related boundary conditions for community resilience is the social, economic, political and environmental/bio-physical context. This includes contextual factors and conditions around the community itself, requiring

the expansion of the analysis of community resilience outward to take into account the wider political and economic factors that directly or indirectly influence the resilience of the community. In different concepts and theories these contextual factors have been addressed, e.g. in institutional analysis (Whaley and Weatherhead, 2014; Ostrom, 2005), common pool resource research (Edwards and Steins, 1999) or socio-ecological systems research (Orach and Schlüter, 2016).

The analysis of contextual factors can also expand backward in time and include an analysis of change over time. Therefore,

apart from the more or less stable context factors we include as another element social, economic, political and environmental change over time as an influencing force of extra-community framing of community resilience. Disaster risk and hazard research scholars (Birkmann et al., 2010) as well as policy change scholars (Orach and Schlüter, 2016) have identified different dynamics and types of change from gradual, slow onset change to rapid and abrupt transformation, from iterative to fundamental changes. This can include social change, economic change and policy change as well as changes in the natural

environment, e.g. connected to climate change and land degradation.

Considering the third boundary condition, a broad variety of disturbances can influence the community and its resilience partly closely interlinked with the perceived or experienced changes and the specific context factors. As already noted by Wilson (2013), disturbances can have both endogenous (i.e. from within communities, e.g. local pollution event) and exogenous causes (i.e. outside communities, e.g. hurricanes, wars) and include both sudden catastrophic disturbances (e.g. earthquakes) as well

as slow-onset disturbances such as droughts or shifts in global trade (for a typology of anthropogenic and natural disturbances affecting community resilience cf. Wilson, 2013). In line with Wilson we conclude that communities are never 'stable' but continuously and simultaneously are affected and react to disturbances, change processes and various context factors. Therefore, disturbances can not only have severe negative impacts on a community but also trigger change and transformation that might not have activated otherwise. As a result, in empirical applications a clear-cut differentiation between contextual

change over time and slow-onset disturbances or disturbances that trigger change is not always possible.





## 5 Discussion and conclusion

### 5.1 Interlinkages between the domains and extra-community framing

Considering the intertwined components of the proposed framework, research can be guided by acknowledging the complexity of the possible interactions between the resources & capacities, learning and actions domains in shaping community resilience.

Therefore, efforts to evaluate these multiple levels; their interactions; and how they operate in different contexts for different hazards can provide an enriching evaluation of community resilience.

An example of how the emBRACE framework of community resilience helped to reveal the interrelatedness of socio-political and human resources in the civil protection actions, and the importance of social solidarity and trust as important contextual factor, is delivered in the case study work in the city of Van, Turkey. Here the exploration of individual resilience after a severe

earthquake proved how influential the contextual factors are. The results indicated that the political context played an important role in shaping survivors' perceptions of their own resilience. Doğulu et al. (2016) shows that community resilience is facilitated when provision of post-quake aid and services is based on equality and trust (and not nepotism and corruption) and not hindered by discrepancy of political views among both government bodies, community members and NGOs.

Further, the analysis revealed that the earthquake experience in the Marmara region of Turkey in 1999, twelve years earlier,

influenced the resilience of the community following the Van earthquake, based on learning processes that resulted, for example, in a change in the public disaster management by state organisations as well as the adoption of new legislation. Thus, especially for the state institutions, the impact of a past disturbance, may lead to significant changes in disaster risk management, which in turn is likely to contribute to fostering of community resilience in Van and beyond (Karanci et al., 2017). This example shows how the framework provides an understanding of the interrelatedness of the three domains and the

importance of their interactions in shaping community resilience. Yet, the specific types of relations and interlinkages are case specific, i.e. influenced by various external variables. To specify these and develop typologies of linkages and relations needs to be investigated in further research.

### 5.2 Application and operationalization of the framework in indicator-based assessments

The emBRACE framework for community resilience was iteratively developed and refined based on the empirical research of

the specific local-level systems within the five case studies of emBRACE, thus is strongly supported by local research findings on community resilience. It was mainly developed to characterize community resilience in a coherent and integrative way. Nonetheless, it was also developed to be applied for measuring resilience and thus a heuristic to be operationalized in form of an indicator based assessment. Thus, the framework provides one possible – but empirically legitimized – structure and route to select and conceptually locate indicators of community resilience.

Within the emBRACE project we derived case study specific community resilience indicators as well as a set of more concise, substantial indicators that are generalizable across the case studies (Becker et al., 2017). The generalizable key-indicators include a wider range of indicators from more quantitative indicators like the presence of an active third sector emergency



coordination body, or the percentage of households in the community subscribed to an early-warning system, operationalizing the domain of civil protection action up to more qualitative indicators such as social/mutual trust and the sense of belonging to a community applying the domain of human and social resources and capacities.

Besides identifying and selecting suitable indicators, it is crucial to understand how to develop, integrate, interpret and apply indicators (Jülich, 2017a; Bahadur and Tanner, 2014). Concrete instructions are needed to provide a useful source of information for proper indicator application in practice and we recommend using some form of guideline for community resilience indicator development (cf. for example Becker et al., 2015). In particular, the possible methods of data collection for the constituent parts of this framework require attention, since they affect not only the methods adopted to parameterise the indicators, but also the scale of application.

**5.3 Reflections on the results and emBRACE methodology and limits of the findings**

The proposed three-layered framework for characterizing community resilience is developed deductively by considering theoretical approaches of resilience from various disciplinary backgrounds and state of the art research: it is also developed inductively based on empirical insights from our case study work. The result is a theory-informed heuristic that guides empirical research as well as disaster management and community development.

Research does not necessarily include all domains and elements but often focuses on some specific domains and their interaction in more detail. When guiding disaster management and community development the framework helps to highlight the importance of the multiple factors that are related to community resilience. Whether the framework informs scientific or more practical applications, in most cases it is necessary to adapt the framework to the specific context to which it is applied, e.g. cultural background, hazard types or the socio-political context.

Nevertheless, it is developed as a heuristic device, i.e. a strategy based on experience and as an aid to communication and understanding, but not guaranteed to be optimal or perfect. Further, the framework should be subject to further research both for further conceptualizing community resilience and applying and specifying the framework in various contexts of community resilience.

**Disclaimer and Acknowledgments**

This study has received funding from the European Union's Seventh Framework Programme for research, technological development and demonstration under grant agreement No. 283201. This article reflects the views of the authors only, and the Commission cannot be held responsible for any use which may be made of the information contained therein. Special thanks go to Astrid Björnsen Gurung and Valentin Ruegg for their graphical support during the framework development.





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




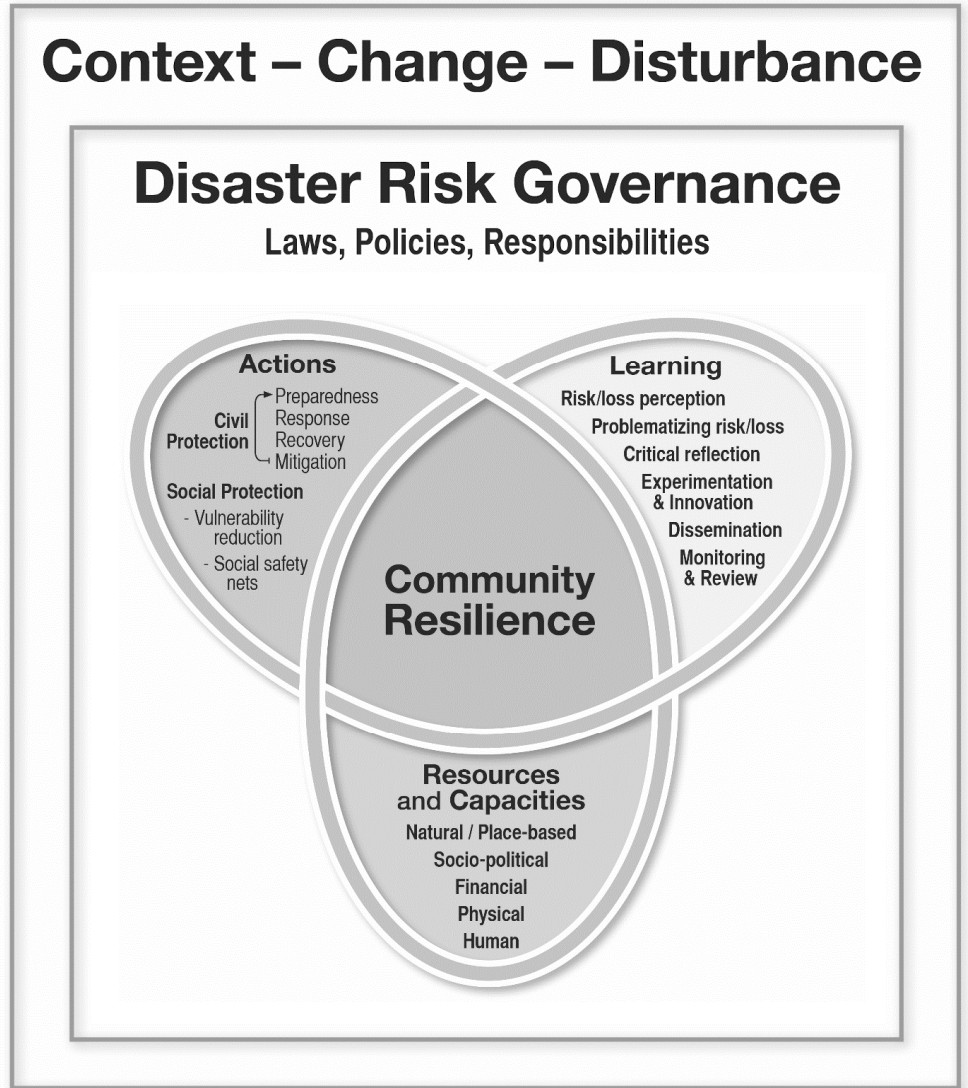

**Figure 1: The emBRACE framework for community resilience to natural hazards (source: own illustration).**