# Peer review of "Conceptualizing community resilience to natural hazards — the emBRACE framework"

_Natural Hazards and Earth System Sciences, 2017_

## Referee Comment (RC1) · I. Kelman (Referee) · 19 May 2017

This paper is well-structured and well-written, covering an important topic which is relevant to this journal. The manuscript displays good use of apposite mixed methods, presenting solid interweaving of theory with empirics. The text is somewhat sparse on many of the methodological details, but this approach to the writing is understandable given the nature of the paper and what it aims to represent. The manuscript is also sparse on verifiable and specific results, but again this approach is understandable given the nature and approach of the paper.

Overall, this paper is not a basic science piece with hypotheses-methods-results-discussion. Instead, it is mainly conceptual while being methods orientated, bringing together a large volume of work which is synthesised into an applicable framework.

The paper is important and successful in this approach, hence is publishable as such. The empirical contribution is especially useful, despite the lack of details, because details are given in other publications cited and because the idea of this manuscript is to meld theory and empirics to develop the framework presented. As such, the empirics contribute what they should to this form of paper.

The main concern with the paper, requiring major revisions, is the theoretical baseline and theoretical approach, leading to questions about the final framework.

The paper explicitly accepts from the beginning that both 'resilience' and 'community' are poorly theorised and are contested concepts. This approach is impressive and helpful, which led to a promising start to the manuscript. Unfortunately, the paper then does not add substance to either concept while perpetuating the retrogressiveness of certain classes of literature which the paper initially (and insightfully) critiques.

For example, for 'resilience' why would the authors 'broadly follow the definition of resilience proposed by the Fifth Assessment Report of the Intergovernmental Panel on Climate Change (IPCC)' when the critical resilience literature points out substantive flaws in this definition and provides ways to improve it? A golden opportunity is available–with the rich data and thorough investigation done in this study–to do better than the IPCC and to build on much better, existing work. The authors do not even cite a seminal paper published in this journal http://www.nat-hazards-earth-syst-sci.net/13/2707/2013/nhess-13-2707-2013.pdf The authors do note a few of the key critical resilience publications, but do not acknowledge those publications' critiques of the IPCC and its definitions.

I can possibly guess why the authors wholeheartedly embrace the IPCC approach: because so many others do so. Which is exactly why this component of the paper lacks substance: because so many others do the same. If we do what everyone else is doing, where is the originality and what is the point? When so many critiques exist of the approach taken, a paper such as this one could be bold, applying the alternative

approaches to the empirical case studies to judge whether or not the critical social science literature on resilience really stands up to empirical scrutiny.

Similarly with 'community', where is the literature by Terry Cannon explaining difficulties with the concept of 'community' in exactly the contexts which this paper covers? Again, I appreciate that the authors acknowledge the theoretical challenges and this is needed. But there is no point in producing a framework when those theoretical challenges are not overcome and are actually reinforced. The point of critical approaches is to do better, not to perpetuate the same difficulties which the authors have explicitly acknowledged.

Even for the statement 'only a few approaches have tried to characterize and measure community resilience comprehensively', which is fair, the authors do not cite the best work available: that of Karen Sudemeier, James Lewis, and John Twigg. When the baseline literature review has missed key material, the paper's aim cannot be entirely fulfilled and, in fact, the paper's approach and framework do not progress the field from what is already published–and published too many times. Conversely, the literature which this paper misses contributes to filling in the gaps which the authors correctly identify and rightly seek to rectify. This situation does not preclude the need for further work, as outlined by this paper, nor does it claim a lack of the gaps which these authors identify. It simply means that no paper can ever be comprehensive regarding citations, so references need to be selected much more carefully than is evident in this manuscript.

Continuing along this same theme, the paper relies on the concept of 'adaptive capacities', but it does not explain how this concept is different or provides any more than the concept of 'capacities'. As it turns out, the concepts which are reasonably well described and quite well applied in the paper are straight from the 'capacities' literature from long before the 2011 citation given–and this manuscript is strong for it. But it is important for any current work to recognise the baseline of literature which came before it. In focusing on adaptive capacities rather than capacities, this paper does not

do so.

There is useful discussion of 'individual resilience' and 'survivors' perceptions of their own resilience'. This material has been extensively discussed in psychology literature (e.g. Johnston, Lewis, Ronan, Paton), none of which is cited in this paper, apart from one Paton reference to a conference piece rather than to his detailed work in peer-reviewed journals. Interestingly, page 2 refers to this cohort of psychology research but no citations are given. Consequently, its rich history and material are absent from the paper, again indicating that citation choice could be improved.

As such, the 'iterative process building on existing scholarly debates' claimed by the paper and the literature review are inadequate. Substantial revisions would be required on the theoretical part of this paper.

Other theoretical components which require further thought and exploration:

(a) Financial and monetary are seen as being synonymous. This assumption ought to be interrogated.

(b) Phrases such as 'civil protection' and 'social protection' have multiple meanings and debates. The 'disaster management cycle' concept has been eviscerated in the literature with alternatives provided. It is puzzling why the cycle approach is presented without recognising why a almost a generation of literature has been explicitly trying to move away from it.

(c) The term 'transformation' appears sporadically throughout the paper, without definition, without critique, and without acknowledging the critiques of contemporary transformation literature. Perhaps it would be easiest not to refer to 'transformation', because a single manuscript cannot cover everything.

(d) The authors could take more care in reading and applying their own writing. As one example, consider 'It is an ever emergent property of social-ecological and technological systems coproduced with individuals and their imaginations'. Why not use your

'imaginations' to explain why 'social-ecological and technological systems' should not be separated and why 'socio-ecological systems' is inherently problematic? This is not the only example of jargon subsuming reality, which would be so easy to avoid.

(e) 'Disturbances' is a term from ecosystem science, imposed on social-ecological systems theory discourse. It is not clear that this word has relevance to society in reality, which is one of the critiques of resilience approaches not discussed in the paper.

(f) There is some, but not extensive, theoretical and conceptual novelty in this paper compared to complex adaptive systems theory (although this paper uses quite different vocabulary for similar concepts in complex adaptive systems theory). This is by no means a defence or endorsement of complex adaptive systems approaches. It is querying why the authors would go to such an extent to produce a framework, much (not all) of which is already in the literature, even with different labels–and which many like and many dislike.

In the end, I could probably be convinced that the ultimate result of this paper does not change much. This is especially the case given that the framework produced is meant to be applied and is applied in the paper. But plenty needs to be done in order produce a convincing theoretical and critiquing pathway to this end result.

Regarding the journal's questions:

1. Does the paper address relevant scientific and/or technical questions within the scope of NHESS? Yes.

2. Does the paper present new data and/or novel concepts, ideas, tools, methods or results? Reasonably.

3. Are these up to international standards? Reasonably.

4. Are the scientific methods and assumptions valid and outlined clearly? No, but many details are provided in citations given or else changes are requested in the main comments.

5. Are the results sufficient to support the interpretations and the conclusions? Reasonably.

6. Does the author reach substantial conclusions? Reasonably.

7. Is the description of the data used, the methods used, the experiments and calculations made, and the results obtained sufficiently complete and accurate to allow their reproduction by fellow scientists (traceability of results)? No, but this is not a concern. These details are provided in citations given.

8. Does the title clearly and unambiguously reflect the contents of the paper? Yes.

9. Does the abstract provide a concise, complete and unambiguous summary of the work done and the results obtained? Yes.

10. Are the title and the abstract pertinent, and easy to understand to a wide and diversified audience? Yes.

11. Are mathematical formulae, symbols, abbreviations and units correctly defined and used? If the formulae, symbols or abbreviations are numerous, are there tables or appendixes listing them? Not applicable.

12. Is the size, quality and readability of each figure adequate to the type and quantity of data presented? Yes.

13. Does the author give proper credit to previous and/or related work, and does he/she indicate clearly his/her own contribution? Reasonably.

14. Are the number and quality of the references appropriate? The number is fine, but not the quality. See the main comments.

15. Are the references accessible by fellow scientists? Yes.

16. Is the overall presentation well structured, clear and easy to understand by a wide and general audience? Yes.

17. Is the length of the paper adequate, too long or too short? Neither too long nor too short.

18. Is there any part of the paper (title, abstract, main text, formulae, symbols, figures and their captions, tables, list of references, appendixes) that needs to be clarified, reduced, added, combined, or eliminated? See the main comments.

19. Is the technical language precise and understandable by fellow scientists? Yes.

20. Is the English language of good quality, fluent, simple and easy to read and understand by a wide and diversified audience? Yes.

21. Is the amount and quality of supplementary material (if any) appropriate? Not applicable.

---

## Referee Comment (RC2) · C. Stephan (Referee) · 28 Jun 2017

This paper presents an interesting and promising framework that is very ambitious in developing an analytical tool and at the same time providing conceptual insight into community resilience. It is an important contribution to conceptual work on resilience and especially community resilience. It is well-written and .provides new ideas and methodological insights to the reader. The need to improve conceptual work on the concepts community + resilience, is identified in an appropriate manner. Key literature that addresses challenges linked to this term is discussed.

A strength of this paper is the connection of the analytical tool and conceptual work. The development of the framework as analytical tool lies at the centre of the work and

is successful. The identification of three domains is a helpful and powerful innovation. However, the challenges that this approach carry along, are not made sufficiently transparent. The vagueness of the concept is not fully overcome. While strands of theory that could help to shed light on the terms are mentioned, they should be made use of in more depth. More conceptual clarity is necessary in order to underline, why e.g. the domain of "actions" has to be approached by a specific type of methodology that encompasses relevant and complex social dynamics. The ambition to connect a top-down systems understanding of resilience and a bottom-up empirical perspective brings some challenges that should be discussed. Specifically the ways in which the three domains are approached methodologically do depend highly on the conceptualisation and operationalisation of the terms.

Social theory is the conceptual source from which new knowledge could be introduced in order to access community resilience. It requires a new approach, maybe refraining from using clear-cut indices, but it is necessary if the social interactions that occur and that could contribute to "resilience" are taken seriously. I propose that this paper puts the open questions that arise from the framework, at the centre and while it is not necessary to answer these questions, it should make the questions as clear as possible in order to give the chance to other authors to respond and to hint towards good conceptual approaches that could help improving the concepts and tools of the embrace framework.

In the following, specific comments to parts of the paper are presented.

Concerning the clarification of the term "community resilience" it would be good to also discuss literature that presents a critical perspective towards the use of the term "community".

Resilience is mentioned as influenced by social interactions, but there is a lack of discussion of social theory that explicitly addresses social interactions and of the contributions it could make to a concept like resilience. As an example, structuralist vs. nonstructuralist lines of theory could be discussed. This could contribute to an improvement of the conceptual approach, especially for the domains of "action" and "learning". A promising approach on understanding resilience in the light of social interactions could be adopted by making use of current developments in social practice theory (authors like Schatzki, Shove but also classics like Bourdieu and Giddens).

In presenting the framework (chapter 4) it would be good to discuss more clearly the types of generalisations that had to be made in order to develop the framework.

The identification of the different capacities and resources does not fully overcome thinking in boxes, e.g. financial, human, natural sector. One could ask whether these are the relevant structuring elements in order to understand COMMUNITY resilience. It is designed in order to build indicators that could be related with figures/numbers, but I doubt that it allows evaluating complex social interactions.

By saying e.g. that civil protection is focusing on hazard specific action, the approach does not look sufficiently into to the multiple social interactions that take place and are of relevance for civil protection actors. It is moreover not clear who performs which types of actions. When talking about Civil protection it is not clear if staff members and their actions only are meant or if it refers to all activities that take place under a civil protection logic independently of who performs these actions.

Learning (and awareness) is mainly explained through a concept of knowledge and thereby it does not address sufficiently the activity and practices that are parts of learning processes. In its current form the domain of learning cannot answer the question why people have more knowledge but do not learn or do not implement the things learned.

The paper makes a relevant point by highlighting that community resilience has to be analysed as part of a broader context. But there is a lack of conceptual depth in dealing with this in this paper. It is not regarded as sufficient to look at what other authors of disaster risk science and policy change have written. This could be improved by

discussing the work on social change from those scholars that analyse the nature of the social itself (social science and humanities).

Concerning the questions to be addressed according to the journal:

1. Does the paper address relevant scientific and/or technical questions within the scope of NHESS? YES 2. Does the paper present new data and/or novel concepts, ideas, tools, methods or results? YES 3. Are these up to international standards? YES 4. Are the scientific methods and assumptions valid and outlined clearly? YES 5. Are the results sufficient to support the interpretations and the conclusions? PARTLY 6. Does the author reach substantial conclusions? YES 7. Is the description of the data used, the methods used, the experiments and calculations made, and the results obtained sufficiently complete and accurate to allow their reproduction by fellow scientists (traceability of results)? PARTLY, see comments above. 8. Does the title clearly and unambiguously reflect the contents of the paper? YES 9. Does the abstract provide a concise, complete and unambiguous summary of the work done and the results obtained? YES 10. Are the title and the abstract pertinent, and easy to understand to a wide and diversified audience? YES 11. Are mathematical formulae, symbols, abbreviations and units correctly defined and used? If the formulae, symbols or abbreviations are numerous, are there tables or appendixes listing them? N.A 12. Is the size, quality and readability of each figure adequate to the type and quantity of data presented? Yes 13. Does the author give proper credit to previous and/or related work, and does he/she indicate clearly his/her own contribution? YES 14. Are the number and quality of the references appropriate? Mainly,but substantial work from social theory should be added. 15. Are the references accessible by fellow scientists? YES 16. Is the overall presentation well structured, clear and easy to understand by a wide and general audience? YES 17. Is the length of the paper adequate, too long or too short? ADEQUATE. 18. Is there any part of the paper (title, abstract, main text, formulae, symbols, figures and their captions, tables, list of references, appendixes) that needs to be clarified, reduced, added, combined, or eliminated? NO 19. Is the

technical language precise and understandable by fellow scientists? YES 20. Is the English language of good quality, fluent, simple and easy to read and understand by a wide and diversified audience? YES. 21. Is the amount and quality of supplementary material (if any) appropriate? N.A.

―――――――――――――――

---

## Referee Comment (RC3) · H. Fünfgeld (Referee) · 30 Jun 2017

This conceptual paper makes an important contribution to ongoing academic discourses on how resilience can meaningfully be applied to the social realm, by providing a conceptual framework for community resilience to natural hazards. The article and the framework presented therein draw on extensive and engaged scholarly work involving in-depth case studies in five countries. The framework combines commonly utilised and readily observable conceptual building blocks, such as different types of community resources and assets, with aspects of discourses on social learning, governance and responsibilisation that have been critically examined in the increasingly vast literature on resilience and its social relevance. To this end, the paper not only makes a meaningful conceptual contribution, it also is of a highly integrative and synthesis-

ing nature that successfully attempts to join up key aspects of otherwise fragmented academic discourse.

Any attempt at characterising an amorphous idea such as resilience is faced with significant theoretical and epistemological challenges. In essence, it is much easier to criticise existing interpretations of resilience and their lack of rigorous theorisation than to come up with an alternative conceptual model that is cognisant of the theoretical and practical challenges of (social) resilience concepts, yet at the same time pragmatic and applicable to different situations. Any such model needs to both capture – and explain – at least some of the conceptual breadth that make resilience so attractive for ubiquitous use in the first place, while also being specific enough to make it more than just a loose (subjective) collection of fragmented conceptual ideas. The framework presented in this paper goes a long way towards this integrative goal, by embedding the relatively well defined and more readily observable domains of actions, learning and resources and capacities within broader contexts and boundary conditions that highlight the role of governance, social, economic and political change, and disturbances.

A further challenge – and arguably a more significant one – is that of doing justice, in theoretical and empirical terms, to the notion of community resilience. Community resilience inherently refers to a collective quality, even though the unit of analysis may be an individual household or person. Hence, the framework needs to achieve two things at once: provide a conceptual frame for a social, relational interpretation of resilience while at the same time critically examine 'community' as that social context to which resilience, with all its strong engineering and ecological connotations (see Davoudi et al. 2012), is being applied. A 'community resilience' framework therefore always is, in part, a transposition of ideas that originated in the natural sciences into the human social realm – an intrinsic challenge that the paper could have discussed in more detail upfront.

In the context of this conceptual transfer into the social realm, epistemological questions arise, such as to what extent do the authors take on a constructivist perspective that highlights and problematises, for example, the coding of power differences in politicised languages and knowledges of resilience? To what extent is a more positivist perspective appropriate in this research context and perhaps inevitable, given the underlying project objective of 'characterizing and measuring resilience of European communities' (p.5)? While the reader gets to appreciate the authors' awareness of different epistemological perspectives through, for example, the discussion of resilience as a normative versus analytical concept, much of this important epistemological reflection and argument seems to be hidden 'between the lines' in the text. A brief, more upfront explanation of the underpinning ontology that has guided the endeavour to characterise community resilience would have been desirable, in my view – especially given that the framework decidedly is about 'community', a term that sociologists in particular have debated, deconstructed and subsequently used to re-theorise emerging social dynamics for several decades. Here, a stronger and more critical review of existing interpretations of the term community, including the associated potential of co-optation of heterogeneous, place-based communities into political agendas under the optimistic and potentially 'homogenising' collective guise of community resilience could place the emBRACE framework on a more solid socio-theoretical footing. This would demonstrate more visibly that the framework is not only informed by (social) theory but also reflective of the ontological and epistemological challenges inherent in any attempt towards measuring and quantifying abstract social constructs.

Despite this, the framework manages well to straddle the fluid boundary between specificity and complexity, within the broader assumption that community resilience is in fact something that is 'knowable' and measurable with empirical social research methods. This balancing act manages to avoid an overly mechanistic ('engineering') interpretation that deterministically reduces resilience to a readily measurable, analytical category of community functioning. At the same time, however, it avoids falling into the trap of relegating community resilience to idiosyncrasy that evades any analytical grasp. This balance is achieved by discerning concrete 'domains' of action, learning and resources and capacities that are bounded by contextual enablers and constraint. Here,

the framework strongly resonates with the Sustainable Livelihoods Framework – a conceptual connection the authors have acknowledged and described in detail.

This integration of individual-subjective and collective-institutional dimensions at community level on the one hand and contextual boundary factors on the other hand provides for a balanced, heuristic approach that leaves plenty of (necessary) room for refinement and adjustment of the framework for application in different social and political settings. If anything, a more detailed description of the two extra-community frames and their respective boundaries would likely help better guide and facilitate the process of adjusting the framework to different contexts.

As a researcher interested in questions of equity and justice, I can't help wondering what happened to the ambition and need stated at the beginning of the paper to shed light on the role of power when analysing community resilience. The paper goes to some length to explain processes of top-down responsibilisation, as a way of governments exerting power over their citizens through a resilience framing. Yet the framework does not seem to provide much-needed guidance on how to examine power struggles inherent in local resilience processes (e.g. in the civil protection actions introduced in the 'actions' domain), beyond a brief discussion of socio-political resources and capacities that seems to reflect similar arguments included in the Sustainable Livelihoods Framework. Given the dominance of unresolved questions of power and politicisation inherent in discourses and practical applications of 'resilience thinking', the framework could add truly innovative ideas from critical social and political theory into contemporary disaster risk management thinking, e.g. by providing more concrete conceptual guidance on power issues; by directly drawing on concepts of power to ascertain who benefits most from social protection measures and why; who is involved in processes of social learning for resilience and who is excluded (and why so); and by highlighting in more detail how the disaster risk governance context itself is a manifestation of power struggles taking place between different levels of government and between governmental and non-governmental actors – often with negative impacts on community

resilience.

Lastly, further information could be included on the relative role that deductive framework development played in creating the framework, as opposed to that of inductive processes used by drawing on case study findings. Such expanded discussion could also include – space permitting – a few more examples from the case studies to back up and illustrate conceptual claims made in relation to the relevance of certain characteristics of the framework.

Overall, this paper makes a timely and comprehensive contribution to emerging thinking on community resilience and, in particular, its intrinsic connection with, and dependency on, multiple spheres of regulatory and decision-making context. I see the framework's key strengths in its ability to broadly guide more specific conceptual approaches for exploring particular aspects of community resilience and in its ability to conceptualise resilience as an evolving quality (rather than a bouncing back to a status quo) in which social learning features prominently as a driver of change. Community resilience as portrayed by the emBRACE framework is inherently about learning and evolution – an idea that reflects recent academic debates and seems entirely appropriate given that communities are constantly in flux and required to adapt to various forms of social, economic and environmental change.

Specific comments:

p.2, l.7: What do the authors see as the main problem here - that community is under-theorised or that there is little guidance on how to measure resilience? These are two separate arguments and both can be refuted on the basis of evidence from the literature (there is ample discussion about 'community', in particular in sociological literature, and there is more and more technical work emerging on 'resilience assessment') - but aren't the critical questions to ask: what is 'community' and can 'community resilience' in fact be measured? The authors seem to inherently assume that there are affirmative, constructive answers to both questions, but even so these are still important and valid

questions to ask as part of such a substantive, conceptual contribution as the one presented in this paper.

p.2, l.12: I can be argued that precisely this definition, as widely used as it is, perpetuates a narrow view of socio-ecological resilience that does not take individual or collective, subjective and wider contextual factors into account.

p.3, l.23: Would be good to state here the authors' goal with regard to disentangling (or otherwise dealing with) the integration of analytical and normative aspects of resilience.

p.4, l.27: Might warrant further discussion: Can resilience be a theory of change? Is this what the framework is trying to be in part? Is this only vaguely specified because resilience is inherently vague and it is thus impossible to come up with a general theory of change for/of resilience?

p.5, l. 12: It is not clear at this point in the text whether, at the time the case studies were implemented, any particular concept of community resilience was applied (given a first sketch was deduced from the literature).

p.7, l. 3ff.: In relation to the discussion of financial and some of the other capacities and resources mentioned, a critical question to ask is: how/why are these capacities and resources particularly relevant for community resilience, as opposed to being essential for sustaining a livelihood per se (as stipulated by the SLF)?

p.7, l.28ff.: I am not sure if the choice of terminology here is optimal – summarising all possible community resilience actions under the two headings of civil protection and social protection. While these terms are commonly used in different countries, this is conceptually problematic in that a focus on protection conveys a top-down desire to focus on maintaining a given status quo rather than viewing resilience as a transformative idea. I see these two categories as somewhat ad odds with the notion of learning. In addition, in my view there is a marked tension between the language of social protection and responsibilisation that the authors may want to address in the

text.

p.12, l.9: Here, more detail would be useful on how the (draft?) emBRACE framework helped unearth the role of contextual factors in relation to individual resilience. In other words, how was the framework (as opposed to other analytical frameworks that take contextual factors into account, such as the SLF) useful in querying these aspects?

p.12, l.21: To what extent is this a feasible and fruitful research agenda? If the types of relations are case specific, does it make sense to develop typologies?

---

## Author Comment (AC1) · 10 Aug 2017

Reply to interactive comment by I. Kelman: We appreciate the in most parts positive evaluation of the proposed framework for characterizing community resilience. We understand the comment of I. Kelman as encouragement of being clearer about what the proposed framework does and does not do. We will build on challenges of community resilience and will add a description of the elements of these challenges the model addresses plus differentiating what is still open to resolve. The aim of this is to offer a structured drawing of the research frontier and a drawing of some empirical work that builds upon this frontier. Much more work and thinking will be needed, beyond the scope of the research presented in this paper. Needs for further elaboration: 1. Include more of the critical discussions of the concepts "community" and "resilience"

[Figure]

as well as the rich history in many disciplines We agree that we need to include more explicitly the historical references of the mentioned disciplines that conceptualized resilience and community in the past as well as discuss the critical discussions of the recent past. However, it would clearly be beyond the scope of the paper to unravel the various (criticial) historical strands underlying, for instance, the resilience concept. We therefore point out to some key papers, such as: Alexander, D. E. (2013). Resilience and disaster risk reduction: an etymological journey. Nat. Hazards Earth Syst. Sci., 13(11), 2707-2716. doi: 10.5194/nhess-13-2707-2013 Brand, F. S., & Jax, K. (2007). Focusing the meaning(s) of resilience: resilience as a descriptive concept and a boundary object. Ecology and Society, 12(1), online. Cannon, T., & Müller-Mahn, D. (2010). Vulnerability, resilience and development discourses in context of climate change. Natural Hazards, 55(3), 621-635. doi: 10.1007/s11069-010-9499-4 Cote, M., & Nightingale, A. J. (2012). Resilience thinking meets social theory. Progress in Human Geography, 36(4), 475-489. doi: doi:10.1177/0309132511425708 Pelling, M. (2010). Adaptation to climate change: from resilience to transformation. London/New York: Routledge. Weichselgartner, J., & Kelman, I. (2014). Geographies of resilience: Challenges and opportunities of a descriptive concept. Progress in Human Geography. doi: 10.1177/0309132513518834 Welsh, M. (2014). Resilience and responsibility: governing uncertainty in a complex world. The Geographical Journal, 180(1), 15-26. doi: 10.1111/geoj.12012

2. Select references more carefully and build on the rich history of the discussion This is of course challenging as we already cite references from various disciplines that the current community resilience discourse builds upon; nevertheless we will more explicitly reference the historical origins as well as selecting more carefully those recent references that critically add upon the discussion. Also see comment by Fünfgeld and Stephan on the same issue.

Reply to interactive comment by H. Fünfgeld: Again, we appreciate the in most parts positive evaluation of the proposed framework for characterizing community resilience.

[Figure]

Needs for further elaboration: 1. Discuss the intrinsic challenge that is connected with the transposition of ideas that originate in natural sciences into the human social realm (C2) We will add a discussion on these challenges that arise especially in the historical origins of the concepts of "resilience", "disturbances" and "transformation". Also see comment by Kelman C4 and C5. 2. Make explicit and reflect ontological and epistemological challenges inherent in the proposed framework, including a stronger and more critical review of existing interpretations of the term "community". We agree that we need to be clearer on the ontological and epistemological frames that are inherent to our proposed framework and to define more clearly the concept of community. See also comment by Kelman on including the recent critical debates on "community" and "resilience" 3. Provide more conceptual guidance on how to examine power issues inherent in local resilience processes This is a relevant point that we have not made explicit enough in the description of the conceptual framework. We will add upon this by building on the differentiation of power over, power with and power through. 4. Specific comments Will be addressed when revising the text.

Reply to interactive comment by C. Stephan: We are grateful the in most parts positive evaluation of the proposed framework for characterizing community resilience and valuable suggestions. Needs for further elaboration: 1. Make the challenges the proposed approach carries along sufficiently transparent We agree that we need to be clearer on the challenges that the approach carries along, e.g. the combination of a deductive theory driven conceptualization with an inductive empirical perspective. 2. Put the open questions that arise from the framework at the center. We agree to be more explicit about the open questions that arise from the framework when it is applied to assess community resilience in different contexts 3. Discuss a critical perspective towards the use of the term "community". See comment by H. Fünfgeld on the same issue. 4. Make use of the current developments in social practice theory and social change theory e.g. for enlarging the understanding of learning and knowledge or for reaching more conceptual depth when discussing the influence of contextual factors. We will consider social theory can further inform the proposed framework e.g. in the

dimension of learning processes and the contextual factors. Specifically we will critically reflect how far the proposed framework enables to assess community resilience understood as social interaction.

---

## Author Response (AR1)

**Authors' response to the reviews**

ruse, S., Abeling, T., Deeming, H., Fordham, M., Forrester, J., Jülich, S., Karanci, A. N., Kuhlicke, C., Pelling, M., Pedoth, L., and Schneiderbauer, S.: Conceptualizing community resilience to natural hazards – the emBRACE framework, Nat. Hazards Earth Syst. Sci. Discuss., https://doi.org/10.5194/nhess-2017-156, in review, 2017.

**Response to interactive comment by I. Kelman:**

We appreciate the in most parts positive evaluation of the proposed framework for characterizing community resilience. We understand the comment of I. Kelman as encouragement of being clearer about what the proposed framework does and does not do. We will build on challenges of community resilience and will add a description of the elements of these challenges the model addresses plus differentiating what is still open to resolve. The aim of this is to offer a structured drawing of the research frontier and a drawing of some empirical work that builds upon this frontier. Much more work and thinking will be needed, beyond the scope of the research presented in this paper.

*Comment 1:*

Include more of the critical discussions of the concepts "community" and "resilience" as well as the rich history in many disciplines

1. *Authors' response*: We agree that we need to include more explicitly the historical references of the mentioned disciplines that conceptualized resilience and community in the past as well as discuss the critical discussions of the recent past. However, it would clearly be beyond the scope of the paper to unravel the various (critical) historical strands underlying, for instance, the resilience concept.

2. *Authors' changes*:

With respect to this comment we revised section 2 fundamentally and added both publications with more critical discussions of the concepts of resilience and of community as well as very recent publications on resilience and community resilience. This help to disentangle the roots of the challenges we detect for a comprehensive resilience framework to respond to.

*Comment 2:*

Select references more carefully and build on the rich history of the discussion

1. *Authors' response*: This is of course challenging as we already cite references from various disciplines that the current community resilience discourse builds upon; nevertheless we will more explicitly reference the historical origins as well as selecting more carefully those recent references that critically add upon the discussion. Also see comment by Fünfgeld and Stephan on the same issue.

2. *Authors' changes*: We addressed this comments specifically in section 2 and added literature on the history of resilience and disaster research (e.g. Holling, 1973, 1996; Kates, 1971), history of the discussion on resilience in social sciences respectively in interdisciplinary debate on resilience research (e.g. Kelmnan et al., 2016; Alexander, 2013). Also consider response and changes made related to H. Fünfgeld's comment 1.

**Reply to interactive comment by H. Fünfgeld:**

Again, we appreciate the in most parts positive evaluation of the proposed framework for characterizing community resilience as well as the proposals for improvement.

*Comment 1:* Discuss the intrinsic challenge that is connected with the transposition of ideas that originate in natural sciences into the human social realm (C2)

1. *Authors' response*: We will add a discussion on these challenges that arise especially in the historical origins of the concepts of "resilience", "disturbances" and "transformation". Also see comment by Kelman C4 and C5.

2. *Authors' changes*: In section 1 we elaborated on the challenges that stem from a translation of resilience from natural to social sciences in the second section of the paper, which provides ground for our approach by highlighting some the conceptual tensions that surround resilience research. In particular, we added a brief discussion of how resilience emerged from earlier equilibrium-centered approaches (engineering and ecological resilience), and how this can be traced all the way to current policy applications of resilience (e.g. in emergency planning), which continue to draw heavily on the idea of "bouncing back". We acknowledge that our focus on community resilience opens a range of questions on the social context in which resilience unfolds. This includes, amongst others, question on the nature of disturbances, the role of human interventions, the boundaries of social systems as well as the role of power struggles. The concept of "transformation" is not further elaborated at this point, as its use in the text is marginal and synonym for "fundamental change". Further defining or critiquing this term, which is neither essential nor central to our argument and concept, would distract from the message of the text.

*Comment 2:* Make explicit and reflect ontological and epistemological challenges inherent in the proposed framework, including a stronger and more critical review of existing interpretations of the term "community".

1.  Authors' response: We agree that we need to be clearer on the ontological and epistemological frames that are inherent to our proposed framework and to define more clearly the concept of community. See also comment by Kelman on including the recent critical debates on "community" and "resilience"

2.  *Authors' changes*: In section 2 we elaborated the described tensions between descriptive, analytical and normative approaches to resilience and related epistemological and ontological challenges. We connected them to the history of the resilience concept and the challenge of transposing a concept from natural to social sciences (see comment 1). Further, we added a more critical perspective on community resilience (also see Kelman's comment 1 and the author's reply & changes made).

*Comment 3:* Provide more conceptual guidance on how to examine power issues inherent in local resilience processes

1.  Authors' response: This is a relevant point that we have not made explicit enough in the description of the conceptual framework. We will add upon this by building on the differentiation of power over, power with and power through.

2.  *Authors' changes*: In section 2 we included more explicitly the question of power issues to community resilience and referenced recent and older works on political power and resilience (e.g. Szerszynski, 1999, Fainstein, 2015: 160; Jerneck and Olsson, 2008; MacKinnon and Derickson, 2013, Olsson et al., 2014; Bahadur and Tanner, 2014). In section 4.1.1 we added with reference to Partzsch 2016 and Allen 1998 a more specific conceptual guidance for the analysis of power dynamics.

**Reply to interactive comment by C. Stephan:**

We are grateful the in most parts positive evaluation of the proposed framework for characterizing community resilience and valuable suggestions.

Needs for further elaboration:

*Comment 1:* Make the challenges the proposed approach carries along sufficiently transparent

1. Authors' response: We agree that we need to be clearer on the challenges that the approach carries along, e.g. the combination of a deductive theory driven conceptualization with an inductive empirical perspective.

2. *Authors' changes*:

Challenges connected to community resilience literature are highlighted and connected with the identified gaps in section 2. Challenges that go along with the chosen iterative deductive and inductive research strategy are discussed in section 3 and 5.

Comment 2: Put the open questions that arise from the framework at the center.

1. Authors' response: We agree to be more explicit about the open questions that arise from the framework when it is applied to assess community resilience in different contexts

2. *Authors' changes*:

The discussion of limits and open questions has been widened; nevertheless we do not want to put the open questions at the center of the paper as it aims to propose a heuristic to assess community resilience. This heuristic of course can and needs to be further questioned and developed by both in further empirical and theoretical research.

*Comment 3:* Discuss a critical perspective towards the use of the term "community".

1. Authors' response: See comment by H. Fünfgeld and reply and changes made on the same issue.

*Comment 4:* Make use of the current developments in social practice theory and social change theory e.g. for enlarging the understanding of learning and knowledge or for reaching more conceptual depth when discussing the influence of contextual factors.

1. Authors' response: We will consider how social theory can further inform the proposed framework e.g. in the dimension of learning processes and the contextual factors. Specifically we will critically reflect how far the proposed framework enables to assess community resilience understood as social interaction.

2. *Authors' changes*:

The reviewer 2 is making a very valid point with the option to consider social theory for the development of community resilience as a concept of social interaction. We have addressed this as far as possible in the changes to section 2. Nevertheless, we only touch upon this potential contribution very briefly when elaborating the concept and critical discussion of 'community'. This is because we argue that first, social change theory and practice theory are broad families of theories that would need to be elaborated first before considering their potential for a coherent community resilience framework; it would go beyond the purpose and scope of this article to contribute to a *social theory* of resilience; and second, and expansion in direction of social theory would not fit to the scope of NHESS as an interdisciplinary but mainly natural science focused journal.

[revised manuscript text omitted]